# Effects of antennal segments defects on blood-sucking behavior in *Aedes albopictus*

Yiyuan Zhou[1,2], Dongyang Deng[1,2], Rong Chen[1,2], Chencen Lai[3,4], Qian Chen[1,2]*

**1** Research Center of Eugenics, The First Affiliated Hospital of Guizhou University of Traditional Chinese Medicine, Guiyang, Guizhou, China, **2** Department of Obstetrics, The first affiliated Hospital of Guizhou University of Traditional Chinese Medicine, Guiyang, China, **3** Microbiology and Biochemical Pharmaceutical Engineering Research Center of Guizhou Provincial Department of Education, Guizhou Medical University, Guiyang, China, **4** Department of Nosocomial Infection, The First Affiliated Hospital of Guizhou University of Traditional Chinese Medicine, Guiyang, China

* cathy97031@126.com

## Abstract

After mating, female mosquitoes need a blood meal to promote the reproductive process. When mosquitoes bite infected people and animals, they become infected with germs such as viruses and parasites. Mosquitoes rely on many cues for host selection and localization, among which the trace chemical cues emitted by the host into the environment are considered to be the most important, and the sense of smell is the main way to perceive these trace chemical cues. However, the current understanding of the olfactory mechanism is not enough to meet the needs of mosquito control. Unlike previous studies that focused on the olfactory receptor recognition spectrum to reveal the olfactory mechanism of mosquito host localization. In this paper, based on the observation that mosquitoes with incomplete antennae still can locate the host and complete blood feeding in the laboratory, we proposed that there may be some protection or compensation mechanism in the 13 segments of antennae flagella, and only when the antennae are missing to a certain threshold will it affect the mosquito's ability to locate the host. Through rational-designed behavioral experiments, we found that the 6th and 7th flagellomeres on the *Aedes albopictus* antenna are important in the olfactory detection of host searching. This study preliminarily screened antennal segments important for host localization of *Ae. albopictus*, and provided a reference for subsequent cell biology and molecular biology studies on these segments. Meanwhile, the morphology and distribution of sensilla on each antenna flagellomere were also analyzed and discussed in this paper.

## Introduction

Mosquitoes are one of the greatest public threats to human beings and are even considered the most dangerous animal on earth [1]. Female mosquitoes require blood meals to complete their oogenesis, during which they release pathogens into the host or become infected themselves [2–5]. At present, spraying insecticides is effective in preventing the spread of mosquito-borne diseases, but the long-term use of chemical insecticides can lead mosquitoes to develop

**Data Availability Statement:** All relevant data are within the paper and its Supporting Information files.

**Funding:** National Natural Science Foundation of China (No. 32060167), Science and Technology

Program of Guizhou Province (No. [2019]1031 and No. ZK[2022]459), Department of Education of Guizhou Province (No. KY[2021]204), Scientific Research Project of Guizhou University of Traditional Chinese Medicine (No. 2018YFL170810521), Science and Technology Research topic of Traditional Chinese Medicine and Ethnic Medicine of Guizhou Province Administration of Traditional Chinese Medicine (QZYY-2021-100). The funders had no role in study design, data collection and analysis, decision to publish, or preparation of the manuscript.

**Competing interests:** The authors have declared that no competing interests exist.

resistance and cause environmental pollution. Therefore, understanding the biological mechanism of selection and host location of mosquitoes and developing a control method to reduce the contact between mosquitoes and hosts are important for mosquito control in the future.

Olfactory cues are still considered to be the main drivers, although it is possible that other types of cues (e.g. visual and thermal) also play a role in host detection by mosquitoes [6]. Mosquitoes use their ultrasensitive olfactory system to capture these tiny chemical cues and identify the type of host they are feeding on [7,8]. Therefore, research on the mosquito olfactory system will help to better understand its mechanism and develop green anti-mosquito products that interfere with its olfactory behavior.

However, how the mosquitoes use their olfactory to locate hosts has been an open question. On the one hand, researchers have been trying to find out which odors influence mosquito behavior by stimulating mosquitoes with different odor molecules and compounds through numerous behavioral studies [9–12]. On the other hand, attempts at cellular and molecular explanations have yielded a great deal of important information [13–17], but the biological mechanism has not been fully explained. This paper combines morphological and behavioral studies to determine the functional regions of olfactory organ of mosquitoes to narrow down the range of research targets at cellular and molecular levels.

Antennae are thought to be the most important olfactory organ in mosquitoes since the number of sensilla distributed on the surface accounts for about 90% [18,19]. In nature, however, like all other animals, mosquitoes may suffer from disasters that result in damage to their antennae. It was also found in lab-bred mosquitoes that some individual mosquitoes did have incomplete antennae, either due to their developmental defects or obvious acquired damage. So, do these mosquitoes with incomplete antennae still have the ability to pinpoint their hosts? Or, whether there is some olfactory protection or compensation mechanism in the 13 antennal flagellomeres that have evolved over a long period so that when mosquitoes encounter disasters, the partial loss of flagellomeres does not lead to the reduction of their main olfactory perception abilities (such as olfactory localization needed by feeding and reproduction).

*Ae. albopictus* is an important vector control object in China and is the standard material for the efficacy evaluation of mosquito-repellent drugs in China. In this study, the role of different antennal flagellomeres in the bloodsucking process of *Ae. albopictus* was studied through behavioral experiments, and the external morphology and distribution of antennal sensilla were observed by scanning electron microscopy. Discussion of results in an attempt to narrow the functional area of olfactory localization in mosquito blood-sucking.

## Materials & methods

### Mosquito rearing for stock propagation

A strain of *Ae. albopictus* used in this study was isolated from the wild in Changsha, Hengyang, Chenzhou, and Yueyang counties, Hunan province, which was obtained from the Center for Disease Control of Hunan Province (China). The colony in our laboratory has been in culture since 2016. Adult female mosquitoes were fed on the blood of anesthetized mouse in regularly. Mosquitoes were maintained at 28±1°C and 70–80% relative humidity, with a 14:10 h. light/dark photoperiod according to a described protocol [20]. Larvae were fed on yeast powder and adults were maintained on a 10% sugar solution. For stock propagation, 4- to 5-d old adult female mosquitoes were allowed to take a blood meal from anesthetized mice to lay eggs. The Institutional Reviews Board of The First Affiliated Hospital of Guizhou University of Traditional Chinese Medicine approved all animal procedures (Ethical Application Ref: 20210054).

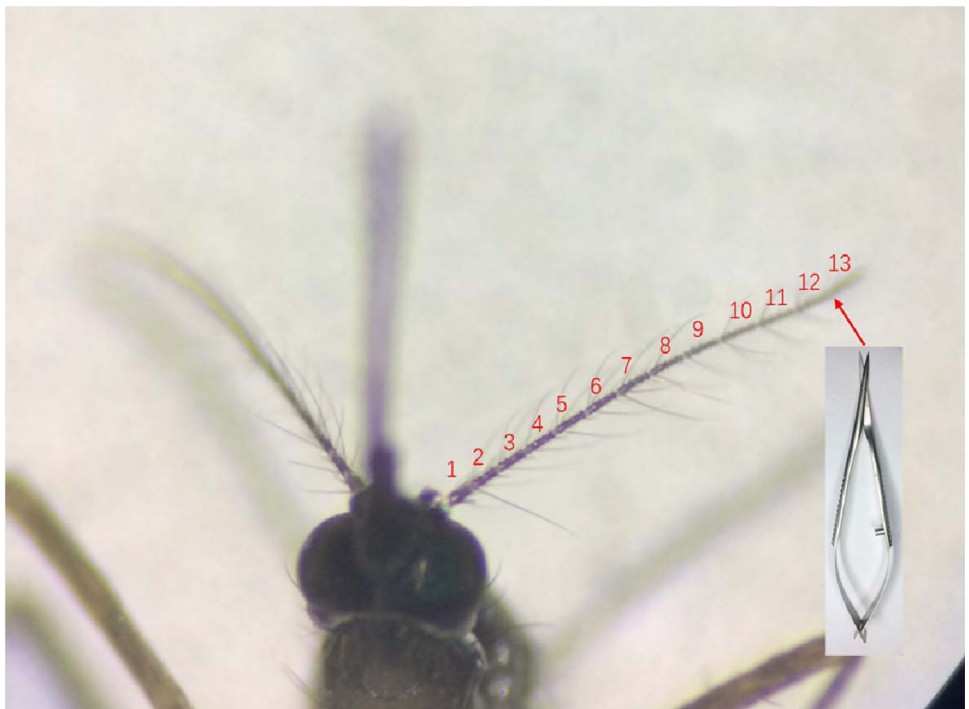

**Fig 1. Schematic diagram of pre-experimental mosquitoes.** In the pre-experiment, the 13th flagellomere segment at the most distal were cut off with Venus scissors.

## Pre-trial for biting assay

Since it is uncertain whether the mosquito will die due to artificial injury when its antennae are cut off, or whether the wound pain will affect its subsequent behavior, pre-trials are needed to determine the impact of antennae-cutting surgery on its life and the healing time of the surgical wound.

4 days after eclosion, 20 female mosquitoes from the same batch were selected randomly, 10 of which were left untreated and fed on 10% glucose water alone as a control group. The other 10 were cold-anesthetized for 1 min and cut off the most terminal section of the mosquito antenna (the 13th flagellomere) with Venus scissors (an ophthalmic surgical instrument) under a stereoscope carefully (Fig 1). Mosquitoes with clipped antennae were put back into the independent cage and fed with 10% sugar water as a pre-experiment group.

Continue to observe and compare the behavior of the mosquitoes in the two cages. 48 hours later, the two groups were deprived of glucose water and fasted for 12 hours. The author put the right hand into the mosquito cage of the control group and the pre-treatment group, respectively, to observe and record the flight and bloodsucking behavior of mosquitoes.

## Biting assay

This assay was based on previously described landing assays [15,21,22], and was designed reasonably as follows to reduce errors.

1. The groups. The behavioral experiment in this paper does not cut out each flagellomere of the antennae one by one but uses a two-step method. First, take 3 flagellomere as a unit, and cut them in sequence to initially lock the range of the flagellomere where the bloodsucking behavior of mosquitoes is significantly reduced. Then, within a locked range of

three flagellomere segments, the flagellomere segments were cut off one by one to further identify the segment of the flagellum that caused the decline in mosquito blood-sucking behavior.

Therefore, behavioral experiments were grouped by the number of missing flagellomere segments, with 10 experimental samples in each group, and 10 biological replicates were arranged to make the total sample size in each group reach 100. Each replicate was derived from the same batch of pupa-emergent mosquito samples, and the samples were age-consistent between biological replicates. To avoid the possibility of accidental death during the mosquito treatment process, resulting in an insufficient final sample size, more than the planned number of mosquitoes in the same batch were selected for antennae shearing, and the living and active mosquitoes were selected for the experiment. Since the blood-sucking behavior of mosquitoes will also be affected at different times of the day, the test sequence between parallel experiments is randomly arranged, that is, parallel experiments are not performed in the order of decreasing or increasing flagellomere segments but randomly shuffled to reduce the influence of test time on the results.

2. The environment. To ensure that the $CO_2$ concentration in the exhaled breath of the volunteers, and the exposed area of the arm are consistent between parallel experiments, each trial was carried out by the same volunteer, wearing the same clothes, making a mark on the right arm 15 cm away from the fist. Whenever the fist was placed in the cage for each assay, the markings on the arms were just at the mouth of the cage to ensure that the arms in the mosquito cages with the same area of the skin surface in each assay. In addition, a platform was fixed 20cm away from the mosquito cage to place the volunteer's head, so as to keep the distance between the exhalation and the mosquito cage constant for each assay (Fig 2).

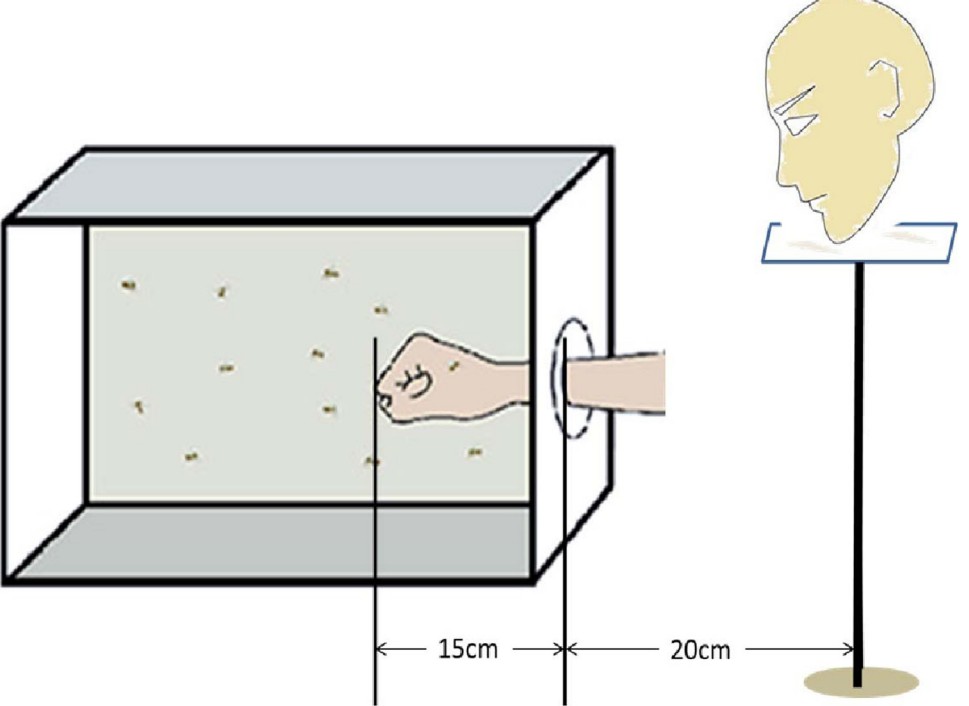

**Fig 2. The schematic diagram for behavioral experiment.**

3. The volunteer. Participants were recruited to participate in the study via flyers on the campus of Guizhou University of Traditional Chinese Medicine on May 27, 2022. A total of 20 students and 3 teachers signed up for this study, which also met the selection criteria as follows: The volunteer was free from chronic illnesses and did not use any medication regularly, and was able to strictly comply with the test requirements which was requested to avoid garlic, onions, alcohol, or spicy food, take a shower using non-perfumed soap every day, and not participate in strenuous exercise before the test. Half an hour before the test, wash their hands with the same perfume-free soap and stop the test immediately if feeling unwell. Given that the study was conducted during the summer vacation (July-August 2022), the final participant in this study was a teacher with the following physical characteristics: Age: 33 years old, Gender: female, Height: 165 cm, Weight: 51 kg, marital status: married without children, attraction to mosquitoes: moderate, non-smoking, non-drinking. She decided to volunteer because she was interested in this study and could tolerate mosquito bites.

4. The mosquitoes. The same batch of adult *Ae. albopictus* were reared together and were given free access to a 10% sucrose solution. 4 days after eclosion, 12 female mosquitoes were selected randomly for each group from the same batch, and cut the antennal flagellomere segments with Venus scissors under a stereomicroscope carefully after a 1 min cold-anesthesia (Fig 3). The cut female mosquitoes were reared in separate mosquito cages in

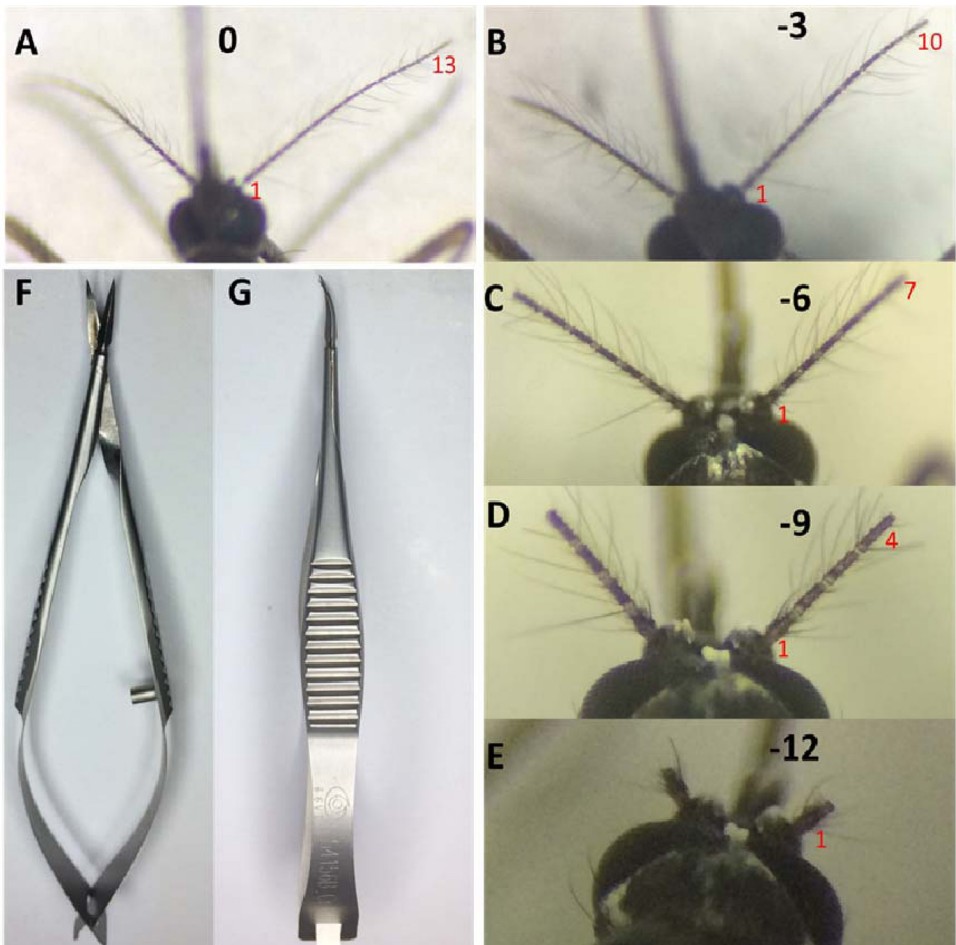

**Fig 3. Antennal artificial cut for the first round blood-sucking behavior test.** A. Antennal without treatment; B~E. 3, 6, 9, and 12 segments of the antenna were cut off respectively; F~G. Venus scissors for cutting antennae.

groups and fed with 10% glucose water for 2 days. Although 12 female mosquitoes treated with antennae were prepared for each group, after fasting for 12 hours, 10 active female mosquitoes were selected for blood-sucking behavior experiments in random order.

5. The Trial. During the trial, the volunteer placed her head on the platform, introduced her arm through cloth sleeves into the cage slowly, and make sure that the mark on her arm was just at the mouth of the cage. Recorded the number of mosquitoes that were blood-fed within 10 min. We defined blood-sucking as landing on the host, inserting the proboscis, and drawing enough blood into the abdomen, or landing on the host, and exploring with the proboscis continuously, which were visible to the naked eye of the observer. After each trial, the volunteers left the chamber for 20 minutes so that the human body odor in the room could be fully dissipated before starting the next trial. Therefore, all trials cannot be completed in one day, but the parallel experiments must be arranged on the same day.

Participants were aware of the content and purpose of the study but were unaware of the test order. This study involving human participants was reviewed and approved in advance by the Institutional Reviews Board of The First Affiliated Hospital of Guizhou University of Traditional Chinese Medicine (Ethical Application Ref: K2022-010). The participants provided written informed consent for participation in this study. The authors had access to information that could identify individual participants during or after data collection. A return visit on 10 September did not suggest that the participant was unwell.

## Scanning electron microscopy (SEM)

Heads with antennae from 4- to 6-day-old adult *Ae. albopictus* were fixed with 4% paraformaldehyde for 2 h at room temperature, after rinsed with PBS (pH 7.3) containing 0.1% Triton X-100 5 times, 50%, 60%, 70%, 80%, 90%, 100% gradient ethanol were used sequentially for dehydration. The samples were sequentially rinsed with a mixture of ethanol and hexamethyldisilazane at the ratio of 75:25, 50:50, 25:75, and 0:100, and then air-dried. The dried sample was glued onto aluminum pin mounts with conductive silver, and gold was sprayed on the surface of the sample in a vacuum sprayer. Samples were observed and digital micrographs of each flagellomere were collected using an S4800 scanning electron microscope (Hitachi, Japan).

## Sensilla counts

Sensillae on each flagellomere were classified and counted by morphology. The average for each sensillum type was calculated for 10 individuals and then multiplied by a factor of 2, assuming only half of the sensilla could be seen in each micrograph.

## Statistical analysis

After completion of the biting assays, the percentage of blood-sucking was calculated by dividing the total number of blood-sucking mosquitoes by the total number of mosquitoes used in each bioassay, multiplied by 100. All statistical analyses were performed by GraphPad Prism 8 software (GraphPad Prism), One-way ANOVA and Bonferroni post hoc analysis were used for group comparisons. Data are presented as the mean ± standard error (SEM) (n = 10). Significance was set at $p < 0.05$.

## Results

### The artificial breakage of the antennae did not cause the death of the mosquito, nor did it affect its activity

The female mosquitoes in the pre-experiment group were all awake after the anesthesia, and there was no death due to antenna breakage. After awakening, mosquitoes will have a short rest (about 20 minutes), and then the flight and feeding behaviors within the next 48 hours were consistent with the control group, showing no listlessness or flight delay. After 12 hours of fasting, mosquitoes in the control group had a strong desire to blood-feed, and the rate of blood-sucking in the control group reached 70% after 10 minutes. While the desire to suck blood was also very strong in the pre-experiment group. Within 2 minutes, 60% of the pre-experiment mosquitoes stayed on the hand and began to suck. After 10 minutes, 50% of the mosquitoes completed blood-feeding, and 30% of the mosquitoes still stayed on the hand to explore and try, the other 20% of the mosquitoes failed to stay on the hand, but they flew for a short time after the hand was introduced into the cage. Although we did not make a statistical analysis, it can be seen that the artificial breakage of the antennae will not cause the death of mosquitoes, nor affect their activities, which was consistent with what we've observed in the lab. After 60 hours of operation, the ability of location and sucking were the same as that of untreated mosquitoes. It can be considered that 60 hours after surgery can be used for a follow-up biting assay.

### The 6th and 7th flagellomere of the antennae of *Ae. albopictus* females may play important roles in their olfactory detection

In the first round with 3 flagellomere segments as the shearing unit, 60 female mosquitoes were divided into 5 groups (Fig 3A–3E), The control group without any treatment (Fig 3A) and the experimental groups in which 3 (Fig 3B), 6 (Fig 3C), 9 (Fig 3D), and 12 (Fig 3E) flagellomere segments were cut off from the end of the antennae, respectively. 60 hours after antennae surgery, mosquitoes with 9 and 12 flagellomere segments defection flew briefly in the cage and moved barely since finding a suitable habitat, only a few of them flew back and forth several times before settling on the volunteer's arm. Through the statistical analysis, the absence of 6 antennae flagellomere segments did not significantly affect the ability of mosquitoes to suck blood, there were still nearly 60% of mosquitoes that can suck blood (Fig 4, see supplementary information). However, less than 10% of them suck blood when the antennae were cut by 9 flagellomere segments. That is, the significant difference in the ability of mosquitoes to suck blood occurred between the absence of 6 and 9 antennae flagellomere segments. And it is speculated that the flagellomere segment affects *Ae. albopictus* to find a host and suck blood may be between or in the 5th and the 7th flagellomere segments.

Then, which flagellomere segment plays a key role? In the next round of experiments, the scope will be narrowed, and behavioral testing will be carried out after cutting out flagellomere segments by section.

According to the above method, a total of 84 female mosquitoes were divided into 7 groups, namely: The control group without any treatment and the experimental groups with 3 (Fig 5A), 6 (Fig 5B), 7 (Fig 5C), 8 (Fig 5D), 9 (Fig 5E) and 12 (Fig 5F) flagellomere segments cut off from the end of the antennal, respectively. 60 hours after treatment, a behavior difference occurred between the 7 and 8 flagellomere segments defection, nearly half of the mosquitoes had blood-sucking behavior could be shown in 7 flagellomere segments defection, but the flying behavior with 8 flagellomere segments cut off decreased significantly (see supplementary information). After a brief period of adaptation, most mosquitoes with 8 flagellomere

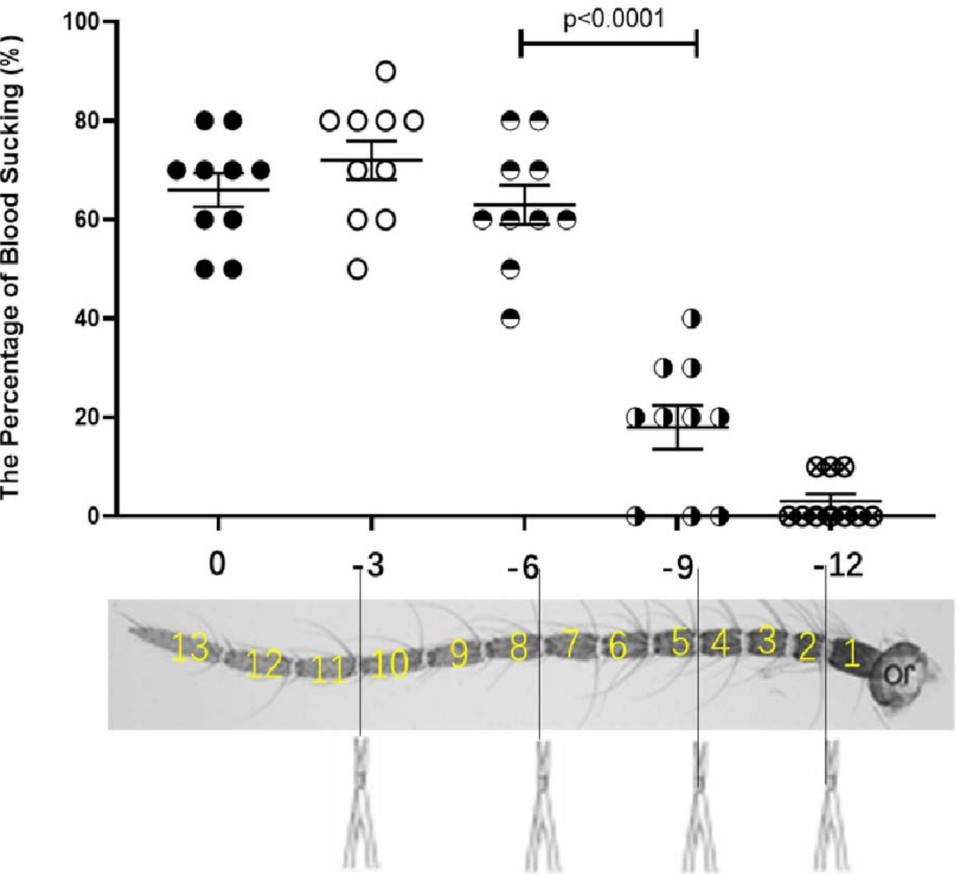

**Fig 4. The percentage of blood-sucking of *Ae. albopictus* with antennal segment defection.** The horizontal axis represents the number of clipped antennal segments, and the vertical axis represents the average blood-sucking percentage. Asterisks indicate significant differences. N = 10 groups, 100 mosquitoes in total.

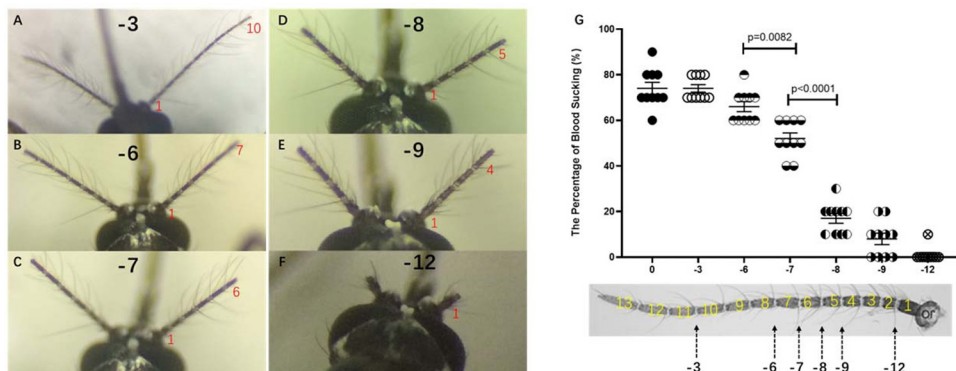

**Fig 5. Antennal artificial cut for the second round blood-sucking behavior test.** A~F. 3, 6, 7, 8, 9, and 12 segments of the antenna were cut off respectively; G. The percentage of blood-sucking of *Ae. albopictus* with antennal segment defection. N = 10 groups, 10 mosquitoes in each group.

segments defection did not move until the trial was over, only a small number of them were seen flitting around in the cage in each replicate, some of which eventually succeeded in sucking blood (only about 10%), while others stayed on the hand without exploring or feeding (Fig 5G). Therefore, in the segments of 4 to 7 antennae, two segments of antennae flagellomere affect blood-sucking behavior, namely the 6th and the 7th flagellomere. That is, when the mosquito has only 6 flagellate segments left (7 segments defection), the blood-sucking behavior is affected to some extent, but when the mosquito has only 5 flagellate segments left (with 8 segments cut off), the blood-sucking ability was lost almost.

## General description of the sensilla of *Ae. albopictus*

The antennal of *Ae. albopictus* female mosquitoes are divided into 13 distinct flagellomeres with a large number of sensilla. According to the description and nomenclature by Zacharuk [23] and Pitts [24], we classified the sensilla by their morphology (Fig 6). The following are detailed descriptions of the types and distributions of female antennal sensilla.

1. Sensilla trichoid: The most widely distributed and most numerous type of sensilla on the antennae with a hair-like structure. According to the shapes, sensilla trichoid were divided into two subtypes: sharp trichodea (sp. trichodea) and blunt trichodea (bl. trichodea) (Fig 6A). sharp trichodea were widely distributed on the antennae, mainly at the 2nd to 13th flagellomeres, and also at the end of the first segment of the flagellum. Blunt trichodea were also mainly distributed at the 2nd to 13th segments of the flagellum, but the number is much smaller than that of the sharp ones.

2. Sensilla chaetica: The longest sensilla with grooved and socketed sturdy bristles (Fig 6). Sensilla chaetica were also divided into two subtypes based on length: large chaetica (lg. chaetica) and small chaetica (sm. chaetica) (Fig 6A, 6B and 6D). The large chaetica were arranged on the basal end, while the small ones were found nearer the distal edge of flagellomeres 2–13. Their numbers decreased from the proximal to the distal flagellomeres.

3. Sensilla basiconica or grooved peg: The sensilla with grooved but no socketed thorn-shaped hair, which is an important feature that distinguishes them from sensilla chaetica (Fig 6B and 6E). The sensilla basiconica was observed on flagellomeres 3–13.

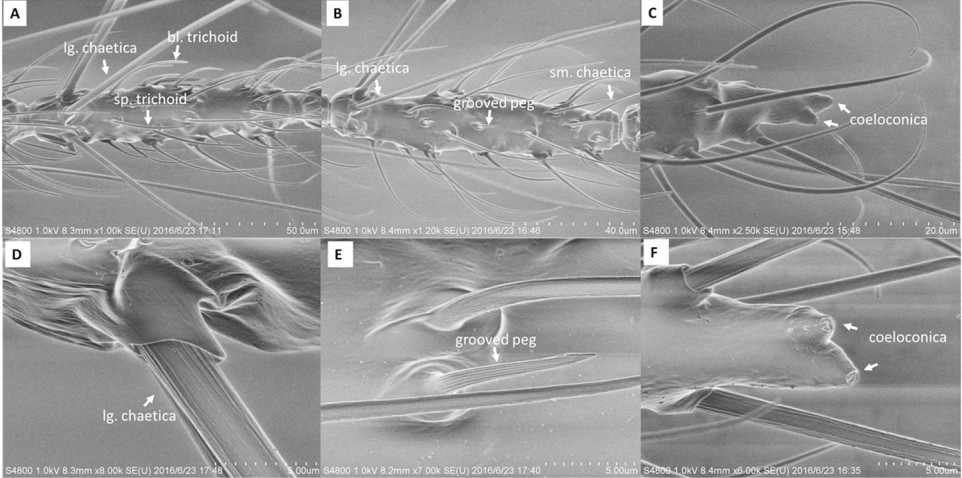

**Fig 6. Sensilla types.** Representative scanning electron micrographs showing sensilla types found on *Ae. albopictus* female antennae.

**Table 1. Types, numbers, and distributions of sensilla (n = 10).**

| Flagellomere | Sensilla trichoid | | Sensilla chaetica | | Sensilla basiconica | Sensilla coeloconica | Total |
|---|---|---|---|---|---|---|---|
| | sp. trichodea | bl. trichodea | lg. chaetica | sm. chaetica | | | |
| 1 | 22 | 8 | 10 | 4 | 0 | 3 | 47 |
| 2 | 32.4 | 4.8 | 6 | 6.8 | 1.2 | 0 | 51.2 |
| 3 | 38.4 | 2.4 | 8.8 | 4 | 4.4 | 2 | 60 |
| 4 | 25.2 | 2.4 | 8 | 2.4 | 4 | 0 | 42 |
| 5 | 33.2 | 3.2 | 7.4 | 2 | 3.2 | 3.2 | 52.2 |
| 6 | 27.6 | 3.6 | 6.4 | 2.8 | 4.8 | 1.8 | 47 |
| 7 | 30.2 | 4 | 7.2 | 3.8 | 4.6 | 2 | 51.8 |
| 8 | 31.8 | 4.6 | 6.2 | 3.2 | 4.8 | 0 | 50.6 |
| 9 | 35.2 | 5.8 | 6.2 | 2.2 | 5.2 | 0 | 54.6 |
| 10 | 35.8 | 5.2 | 6.2 | 2.4 | 4.6 | 2.1 | 56.3 |
| 11 | 38.2 | 4.2 | 7 | 3.2 | 5 | 0 | 57.6 |
| 12 | 37.6 | 4.8 | 6.4 | 3.2 | 5.2 | 1.4 | 58.6 |
| 13 | 42.2 | 8.6 | 6.8 | 1 | 8.6 | 4 | 71.2 |
| Total | 429.8 | 61.6 | 92.6 | 41 | 55.6 | 19.4 | 700.1 |

4. Sensilla coeloconica: The smallest sensilla with pitted pegs (Fig 6C and 6F). The number of such sensilla was also small, they were always found on flagellomeres 1–7, and on the distal tip of the 13th flagellomere with 2.

5. Finally, considering that there might be a large number of certain sensilla arranged on flagellomere 6 and 7, which indicated playing important roles in biting assay, the numbers of each sensillum on each flagellomere were used to find some specials (Table 1). However, unfortunately, we saw no evidence of a distinct distribution pattern of different sensilla types on flagellomere 6 and 7.

## Discussion and conclusions

### Cues for mosquitoes host-seeking

The process of Mosquito host-seeking involves chemical and physical cues [25]. Chemical cues released in host respiration, skin metabolites, and urine, and visual cues may be particularly valuable for a long-distance host orientation, while host physical and chemical cues such as heat, humidity, trace odorant and low volatile may affect host-seeking, landing and probing at proximity. Mosquitoes are believed to detect potential hosts and activate their flight behavior through sparse carbon dioxide plumes at distances of 10 meters or more. When approaching, they mainly rely on visual information and odor cues to help them determine the approximate location of the host. By detecting heat and humidity, they ultimately land on the host, and after landing, their sense of taste will detect the host's skin to find a suitable location for blood-sucking [8,26,27]. The eyes of mosquitoes have relatively poor acuity but high sensitivity to light [28]. Color affects the response of mosquitoes to a certain extent, and they can also determine the direction of flight by comparing shapes [29,30]. In a two-choice test in a recent study, the source of host skin had the highest valence for landing, followed by a combination of heat and visual cues [31]. For several decades, odors have been widely recognized as a critical cue that signals the presence of a host to mosquitoes. Researches on mosquito control strategies have largely focused on the chemical ecology of mosquitoes [26,32–34].

## Mosquito sensory systems

1. Sensory organs. It can be seen that the behavior of mosquitoes in locating their hosts is complex, and involves multiple sensory stimuli. To properly recognize and process these sensory stimuli, mosquitoes have evolved an extraordinary sensory system. Antennal, maxillary palps, proboscis on the head, and tarsi on the legs are all sensory organs of mosquitoes, and they are all multi-sensory integrations. For example, antennae are mainly responsible for acquiring olfactory information but also can sense temperature and humidity [35]. Maxillary palps, in addition to previously reported chemosensory function (including carbon dioxide detecting), were considered potentially involved in mechano-sensation and thermos-sensation [36,37]. Proboscis considered a gustatory organ in food intake was postulated to detect low volatile host cues at proximity, as well as a sense organ of temperature and humidity [13,32]. Tarsi located on the legs of mosquitoes not only respond to external mechanical stimuli, but also have sugar-sensitive hairs like those on the labella, which leads to active sugar-feeding behavior [38–40]. It suggests that mosquito host location involves multiple sensory systems acting together. However, the antennae are considered to be the most important sensory organs in the mosquito host location. This study is also based on the laboratory observation that mosquitoes with incomplete antennae can still locate the host to complete blood feeding, indicating that the remaining antennal segments and/or other sensory organs may play a compensation role in helping mosquitoes complete blood feeding when the antennae are incomplete. So, the question is, to what extent does the absence of antennae prevent them from feeding even though the other organs are healthy? And what proteins are expressed in the missing antennae and what important clues do they perceive? Therefore, this study focuses on the antennae of mosquitoes.

2. Sensilla. Female antennae are thought to play a major role in host detection due to the distribution of approximately 90% of chemosensillar, which are innervated by one or more olfactory receptor neurons (ORNs) whose dendrites extend into the sensillum lymph, while axons project into the main olfactory center in brain, the antennal lobe (AL) [35]. This makes these sensilla the physical sites of chemical detection. The sensilla distributed on antennal were classified into five types by morphology: sensilla cheatica, sensilla trichodea, grooved peg, sensilla coeloconica, and sensilla ampullaceal [24], which are considered generally well conserved among mosquito species. However, The same in a previous study in *Ae. Albopictus* [41], sensilla ampullacea which are considered as probable sites of temperature detection has not been observed in this study [42]. Another study on *Ae. albopictus* divided the sensilla into seven types, namely, addition to sensilla cheatica, sensilla trichodea, grooved peg, and sensilla coeloconica, sensilla auricillica, bohm hair, and sensilla squamiformia were observed. The author mentioned that they did not observe sensilla ampullacea either, probably because they were too few in number or too similar in shape to sensilla coeloconica [43].
Electroantennography (EAG) and Single Sensillum Recordings (SSR) alone or coupled with gas chromatography (GC) were used to detect the responses to compounds of antenna and sensilla. The sensilla trichodea and the grooved peg are sensitive to the attractant scent of the host and play a role in the behavior of the blood-sucking mosquito to find and attack the host. The grooved peg is also sensitive to water vapor, the sensilla coeloconica and the sensilla ampullacea are both thermosensitive sensilla, and the sensilla cheatica is a mechanosensilla, which can sense the movement of air, and plays a role in the upwind directional flight of female mosquitoes against air streams containing the scent of the host [44].

There were significant differences in the type, quantity and distribution pattern of sensilla between the males and females, between culicine and anopheline mosquitoes, and even greater differences between blood and nectar feeders [18,24]. In order to investigate the correlation between the olfactory sensilla and host-seeking behavior, the type, number, and distribution of sensilla in 13 flagellomeres were compared in this study, and the results showed similar distribution and density of sensilla in each segment, especially in the 6th and 7th segments, which showed an important role in behavioral trials, suggesting that other factors may be more important for olfactory driven host-seeking behavior.

3. Cells. Concerning the cellular type repertoire, ORNs are considered to be the most important cell type involved in mosquito olfactory production, as their dendritic membranes express various olfactory-related receptor proteins that can directly interact with odor molecules in the lymph. ORNs then convert these chemical signals into electrical signals and transmit them to higher brain centers [45,46]. Therefore, SSR, which is used to detect olfactory signals in sensilla, can also be used to evaluate the olfactory perception function of the ORNs in them [47]. In addition to the well-known ORNs, a lesser-explored group of non-neuronal auxiliary cells that are adjacent to their respective ORNs are also present in the insect olfactory system, including primordial, primordial, and primordial. These types of cells were initially believed to be involved in the development of sensilla and regulate the ion composition of sensilla lymph [48,49]. In recent years, increasing evidence has shown that many auxiliary cells play important roles in regulating sensory neuron activity, transmission, and structural integrity [50–55]. In mosquitoes, the signal derived from ammonium transporters (Amts), which promote ammonium transmembrane transport and regulate antennal and behavioral responses of Anopheles gambiae, is not only observed in the sensory neurons of the ammonia-responsive basiconic and coeloconic sensilla, but also in non-neuronal auxiliary cells [56]. It suggests that we should pay more attention to the important role of auxiliary cells and their proteins in olfactory sensilla in olfactory perception and olfactory behavior.

## Odor-induced behavioral studies in mosquito

GC coupled with olfactometer or wind tunnel bio-assay were used to determine the effects of each component and its mixture on the olfactory behavior of mosquitoes. Among more than 300 odor components identified by GC-MS, carbon dioxide, L-lactic acid, ammonia, 1-octene-3-alcohol and some carboxylic acids were found to affect the olfactory behavior of mosquitoes [11,57]. Among them, $CO_2$ plays a synergistic role with other host-species volatiles in the host-seeking process of mosquitoes [58], while lactic acid, as a signature human odorant for mosquitoes, has different attractant effects on different mosquito species after coupling with different components [59–61]. Ammonia combined with different components can also have different effects on mosquito olfactory behavior. For example, when ammonia is mixed with lactic acid, it has a synergistic and attractive effect, but when ammonia is mixed with carbon dioxide, it weakens the attraction of mosquitoes [61]. In addition to these generally synergistic odors, some odors have been identified for different mosquito species as potential attractants or attraction inhibitors in mosquito control. For *Ae. albopictus*, a range of saturated acids, unsaturated acids, alcohols, various analogs, and mixtures based on human skin emanations were used to determine flight orientation and electroantennogram response [62–65]. *Ae. albopictus* showed different dose-dependent patterns for different compounds and showed different interests in different combinations. Thus, apart from other physical factors (temperature, humidity), the mechanism in attraction of human odor to mosquitoes is just complex. Much

work remains to be done to fully understand the nature of mosquito host locating on a chemical level. Studies on the cellular and molecular levels of the mosquito olfactory system are also needed.

## Molecular biology studies

In the olfactory organs mentioned above, the occurrence of olfactory involves the involvement of a series of proteins: the odor-binding protein (OBP), which binds to and transports odor molecules in lymph fluid, the olfactory receptors, including Odorant Receptor (OR), Ionotropic Receptor (IR) and Gustatory Receptor (GR), which specifically recognize odor molecules in the dendrite membrane of olfactory neurons, and the odorant-degrading enzymes (ODEs) play a termination role in odor-based signal transduction [14]. Crispr-Cas9-based knock-in strategies were used to label the expression of these proteins to explore the coding patterns of odor, while RNA in situ hybridization, immunofluorescence, and single core RNA sequencing (SNNA-Seq) were also used to locate endogenous expression of these proteins in the sensory system [66]. Cloning and de-orphaning are common methods to identify the olfactory recognition function of these proteins [15,17,67–69], RNAi technology has also been used to identify the effects of these proteins on olfactory behavior in mosquitoes [70,71]. An empty engineered neuron of the Drosophila melanogaster was used as well to express mosquito olfactory-related genes and the responses of this neuron to individual odors were assayed using SSR [72]. Over the past decade, real-time qPCR, transcriptome based on RNA sequencing, and genome-wide analysis of the entire repertoire of olfactory-related genes were widely used to explore a comprehensive expression atlas specific to blood meal, sex, tissue and mosquito species, several olfactory-related genes associated with blood-seeking behavior in mosquitoes have been screened, and the functions in the behavior of which remain to be clarified one by one [73–78].

For *Ae. albopictus*, a large complete genome sequence and transcriptome data were obtained in 2015 [20]. Lombardo provided a detailed transcriptome of the main sensory addendages (antennae and maxillary palps) in *Ae. albopictus* in 2017 [79]. Researches on olfactory-related proteins are ongoing. Genes encoding OBP in *Ae. albopictus* have been identified, and some OBPs with strong female/male expression ratio conducted in the quantitative analysis were considered to be involved in host-seeking of female mosquitoes [80], EAG and behavioral tests were used by Chen to identify OBP which involved in olfactory reception [81], Chen also found that the reduction of Orco transcript levels in *Ae. albopictus* led to a significant decrease in host-seeking and confusion in host preference [82], Some cloned and de-orphaned ORs related to odor-induced behaviors were considered to be used as potential molecular target in mosquito control strategies [83,84]. The expression levels and transcriptional products of IRs in antennae of blood-sucking and non-blood-sucking females were measured in our previous study [16].

Although less research has been done on *Ae. albopictus* than that on *Aedes aegypti* and *Anopheles gambiae* the olfactory behavior and sensilla morphology of *Ae. albopictus* have been described in many Chinese literatures, *Ae. albopictus* also received increasing attention as one of the top 100 invasive species in the world. *Ae. albopictus* is a zoophilic species, but it is highly anthropophilic in nature, visual and chemical cues were used to find its host. It is resistant to commonly used larvicides and adult insecticides, and developing new control tools with proven epidemiological implications is a challenge [85]. Locating the host is a complex process for mosquitoes, and despite ongoing research, there is still no effective way to prevent and control mosquitoes by blocking their sense of smell. The idea of this paper comes from the observation that mosquitoes with incomplete antennae can still successfully suck blood, and we want to know how much antennae loss can affect its behavior. We know that the loss of part of

the antenna does not simply equate to the loss of part of the sense of smell, but may also be related to the loss of temperature, humidity and mechanical sensitivity of the antenna. Our study found that mosquito behavior significantly decreased with the loss of 6~7 antennae segments, a further exploration of which genes are missing in this process that lead to the behavioral changes may hopefully explain the comprehensive factors of mosquito host localization.

Although we designed a simple antennae resection experiment, we still tried our best to control variables to ensure the accuracy of the behavioral experiment. The following explanations are required for the pre-trial in this text.

1. About the shearing position of the antennae in the pre-trial. We chose to cut off only the most terminal segment of the mosquito antenna due to the only segment missing has the least impact on its blood-sucking behavior theoretically. Therefore, the blood-sucking behavior can be used to predict the healing time of the antennae. However, if the antennae of mosquitoes were randomly cut off and the pre-trial was conducted, there would be differences in blood-sucking behavior. It is difficult to determine whether these differences in behavior are caused by the absence of the antennal flagellum segment or the unhealed wound.

2. About the wound. Similar studies were conducted to ablate the antennae, maxillary palps, proboscis and frontal tarsus of mosquitoes separately by microscissor/sharpened tweezers to determine the role of these organs in the sensation of substances. The authors of these studies believed that the effects of treatment on mosquito survival and flight activity during host-seeking behavior were negligible [86,87]. Other studies on Drosophila melanogaster have also employed antennal excising. The authors aimed to expose various cell types in the drosophila antennae by an ex vivo antennal preparation so that live cell imaging can be used to observe the response of these cells when the antennae were stimulated [50,88–90]. Our study focused on the effect of antennal segments loss on insect behavior. Although antennal surgery does not affect the response of cells to stimuli, the pain caused by antennal surgery may affect insect behavior. Therefore, a pre-trial before the biting assay was conducted in this study.

3. Regarding the determination of time for wound healing. Instead of a series of biting assays at consecutive time points, 60 hours after the operation was selected directly. This is because once the mosquitoes have finished blood-feeding, the next physiological process, digestion and egg laying were started, which led difficult to use the same batch of females for continuous time points biting assays. However, using multiple batches of female mosquitoes to conduct biting assays at different time points requires a large workload. Therefore, we chose to give the mosquitoes a longer healing time, and then conducted the pre-trial directly. The results showed that 60 hours after the operation, there was no behavioral effect due to pain. In addition, both the mosquito grouping and the test environment were designed in the final biting assays to reduce the experimental error.

In order to understand the attractiveness of humans tomosquitoes, antennal sensilla have previously been described and compared for several mosquito species. Considering the 6th and 7th flagellomere of the antennae of *Ae. albopictus* females may play a key role in their blood-seeking behavior, it is possible that variations in types or numbers of sensilla exist between them and other flagellomeres, which may suggest areas of future investigation. As such, a comparative examination of the sensilla of each flagellomere on *Ae. albopictus* female antennae was conducted. However, the *Ae. albopictus* females carry the same morphological types of sensilla and the densities of each type are effectively equal between the 6th and 7th flagellomere and other antennal flagellomere. And this result was similar to the result of Qiu [91], who identified 6 functional groups of trichoid and 5 functional groups of GP on segments

6–13 of the antenna based on the responses to a panel of 44 compounds in *Anopheles gambiae* by Single-sensillum recordings. The study has shown that the 13th segment distributed the most functional sensilla, which is probably because it is the first part to be exposed to odors, allowing mosquitoes to respond quickly. In the case of the antennae damage, the abundant sensilla on the most distal segment are missing, but the sensilla distributed on other segments of the antennae can be replenished. Qiu's test indicated that the sensilla on segments 6 and 7 of the antennal flagellum were mainly responsive to some carboxylic acids, alcohols, phenols, ammonia, and amines. Since the authors focussed on segments 6–13 of the antenna, we did not know whether these functional sensilla were also distributed on the segments 1–5.

## Shortcomings and limitations

The limitation of this study are (1) the time for wound healing we used: 60 hours after the operation was based on the absence of the most terminal segment of the mosquito antenna, in which the localization and detection abilities of mosquitoes were almost the same as those of untreated mosquitoes. With the increase of missing antennae, the flight activity, orientation speed, and probing time of mosquitoes decreased gradually according to our observations. Thus, could the reduction of these abilities due to the fact that the closer the cutting position is to the base, the thicker the antennae, and the less adequate the wound healing? The larger the wound, the greater the probability of infection. After all, we did not do anything to seal the wound. In other insects, the cutting of sensory organs incurs some degree of communication dynamics disruption, and laser sealing is the common method for wound sealing [92]. It would be better if there could be a comparison between the antennae cut group and the post-cut laser sealing group. (2) We also did not study the orientational activity of the antenna. It will be meaningful to observe and record whether the remaining antennae still exhibit orientational activity after being severed, especially in the absence of the 6th and 7th flagella, which we found resulted in behavioral differences. (3) If more different volunteers had joined the experiment to increase the selectivity of mosquitoes, would we have gotten different results? (4) As mentioned above, different sensory organs play different roles at different distances. Our results are limited to explain what happened in our experimental space, but this method can also be used to test other distance ranges.

Of course, the response of different parts of the antennae to odors has been studied in other insects for a long time, using methods that are more intuitive than the ones we used in this study. For example, live cell cation imaging was used to observe the response of cells in the isolated antennae. For mosquitoes, however, although many factors influencing their behavior have been identified at the level of compounds, cells, and molecular repertoire, the mechanism by which mosquitoes locate their hosts remain poorly understood [7], and larger-scale response studies have been limited by the array of mixed ORNs on antennal. Therefore, this study aims to first screen out the range of antennae segments causing behavioral differences through relatively simple and extensive behavioral methods, to provide a reference for further researches through cell biology and molecular biology, and also the first step for us to make more meaningful work in the future. Based on this, we are further detecting and comparing the ORNs-related proteins expressed on each segment (as well as those proteins associated with non-neuronal helper cells), and we are still working on it despite the difficulties.

## Supporting information

**S1 File. Raw data of mosquito blood-sucking behavior experiment.**
(XLSX)

## Author Contributions

**Conceptualization:** Qian Chen.

**Data curation:** Rong Chen.

**Formal analysis:** Rong Chen.

**Funding acquisition:** Qian Chen.

**Investigation:** Yiyuan Zhou.

**Methodology:** Qian Chen.

**Project administration:** Yiyuan Zhou.

**Resources:** Yiyuan Zhou.

**Software:** Dongyang Deng.

**Validation:** Dongyang Deng.

**Visualization:** Dongyang Deng.

**Writing – original draft:** Yiyuan Zhou, Rong Chen.

**Writing – review & editing:** Chencen Lai, Qian Chen.

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
