## [Decision Letter · Decision Letter 0]

3 Nov 2022

PONE-D-22-26785Effects of antennal segments defects on blood-sucking behavior in Aedes albopictusPLOS ONE

Dear Dr. Chen,

Thank you for submitting your manuscript to PLOS ONE. After careful consideration, we feel that it has merit but does not fully meet PLOS ONE’s publication criteria as it currently stands. Therefore, we invite you to submit a revised version of the manuscript that addresses the points raised during the review process.

We look forward to receiving your revised manuscript.

Kind regards,

Jiang-Shiou Hwang, Ph.D.

Academic Editor

PLOS ONE

Journal Requirements:

"National Natural Science Foundation of China (No. 32060167), Department of Education of Guizhou Province (No. KY[2021]204), Guizhou Science and Technology Department (No. [2019]1031 and No. ZK[2022]459), Scientific Research Project of Guizhou University of Traditional Chinese Medicine (No. 2018YFL170810521)."

4. Thank you for stating the following in the Funding Section of your manuscript: 

"This work was supported by the National Natural Science Foundation of China (No. 32060167), Department of Education of Guizhou Province (No. KY[2021]204), Guizhou Science and Technology Department (No. [2019]1031 and No. ZK[2022]459), Scientific Research Project of Guizhou University of Traditional Chinese Medicine (No. 2018YFL170810521)."

"National Natural Science Foundation of China (No. 32060167), Department of Education of Guizhou Province (No. KY[2021]204), Guizhou Science and Technology Department (No. [2019]1031 and No. ZK[2022]459), Scientific Research Project of Guizhou University of Traditional Chinese Medicine (No. 2018YFL170810521)."

Reviewers' comments:

Reviewer's Responses to Questions

**Comments to the Author**

1. Is the manuscript technically sound, and do the data support the conclusions?

Reviewer #1: Yes

Reviewer #2: Yes

2. Has the statistical analysis been performed appropriately and rigorously? 

Reviewer #1: Yes

Reviewer #2: Yes

3. Have the authors made all data underlying the findings in their manuscript fully available?

Reviewer #1: Yes

Reviewer #2: Yes

4. Is the manuscript presented in an intelligible fashion and written in standard English?

Reviewer #1: Yes

Reviewer #2: Yes

5. Review Comments to the Author

Reviewer #1: It is an interested study and provided some interested findings for further understanding the antenna with sensors /receptors function for continuing blood feeding. The results are not surprised because the antenna with sensors is for a long range purpose to find the odor and hosts. There are many studies on the host-seeking and blood feeding behaviors of Aedes albopictus in China and other Asian countries in the 1980's and 1990's. The host-seeking and blood feeding is a complicate issue, not only the receptors on the antenna, also, other parts of organs, such as papa, proborese , and tarsal, mating status, distance between hosts and mosquitoes, and other many factors influence on the host-seeking and blood- feeding behaviors. The following questions and comments may assist the authors for further revision:

1. Provide the strain of Aedes albopictus (Hunan?) and more information about the laboratory colony (how long of the colony you and they have and what kind of regular blood resource used even if this species is an opportunity feeder).

2. Have you sealed the antenna after you cut? What was the speed for orientation, probing time, and how long for the blood engorgement, compared with non-cut individual mosquitoes?

3. Provide more details about human volunteers due to the variation by individual.

4. Italic for all scientific name of the species in text and references.

5. More discussion about the receptors/sensors. There were many studies about the olfactory sensors, also, single sense cell function/role detected by EAG ..

6. There were several publications about the receptors of antenna of Aedes albopictus in China. You need to search for the Chinese literatures and understand where you stand and start. Huang, XJ, et al. 1991. Studies on the ultra-structures of antennal receptors of Aedes albopictus. Chinese Journal of Vector Biology and Control Vol. 2 (suppl.): 29-32. Xue RD. 1991. Studies on the olfactory behavior of Periplaneta americana, Musca domestica, and Aedes albopictus (NSF project summary). Chin J Vector Biol Control. Vol. 2 (suppl):5-9. Also, there is a reviewing article about this subject. Xue RD. 2016. Host-seeking and blood feeding behavior of Aedes albopictus. Technical Bulletin of the Florida Mosquito Control Association, Vol. 10:2-13. You may find this article from the ResearchGate or the Florida Mosquito Control Association website at www.yourfmca.org

Reviewer #2: Nice research, good interpretation of data. However, two points: One, this is not frontline research. I have attached a file the stand of the research of today. With the molecular tools we have today, we should use them to get a clear idea where we could interact genetically. The field is so hot these days that it is hard to keep up. However, that is the name of the game. Two, by cutting the antennae short one does cut all sensory system short. To attribute a change of behavior to olfaction may not be correct. One would have to show that mechanoreception has not been altered.

I can see that a more thorough literature review would improve the manuscript. For example, similar research has been conducted with other small animals (e.g., zooplankton, fruitflies) with similar results. This would then give a better platform for their observed data.  major edit

6. PLOS authors have the option to publish the peer review history of their article (what does this mean?). If published, this will include your full peer review and any attached files.

Reviewer #1: **Yes: **Rui-De Xue

Reviewer #2: **Yes: **J R Strickler

---

## [Author Response · Author response to Decision Letter 0]

8 Dec 2022

We thank the editor and reviewers for their time, feedback, and constructive suggestions. We have addressed the reviewers’ comments point by point below. Changes have been documented in the manuscript in highlight text. We have also added a few citations to the manuscript.

Reviewer #1: It is an interested study and provided some interested findings for further understanding the antenna with sensors /receptors function for continuing blood feeding. The results are not surprised because the antenna with sensors is for a long range purpose to find the odor and hosts. There are many studies on the host-seeking and blood feeding behaviors of Aedes albopictus in China and other Asian countries in the 1980's and 1990's. The host-seeking and blood feeding is a complicate issue, not only the receptors on the antenna, also, other parts of organs, such as papa, proborese , and tarsal, mating status, distance between hosts and mosquitoes, and other many factors influence on the host-seeking and blood- feeding behaviors. The following questions and comments may assist the authors for further revision:

1. Provide the strain of Aedes albopictus (Hunan?) and more information about the laboratory colony (how long of the colony you and they have and what kind of regular blood resource used even if this species is an opportunity feeder).

A strain of Ae. albopictus used in this study was isolated from the wild in Changsha, Hengyang, Chenzhou, and Yueyang Counties, Hunan Province, which was obtained from the Center for Disease Control of Hunan Province (China). The colony in our laboratory has been in culture since 2016. Adult female mosquitoes were fed on the blood of anesthetized mouse in regularly. We have added this to the section of “Mosquito Rearing for Stock Propagation”

2. Have you sealed the antenna after you cut? What was the speed for orientation, probing time, and how long for the blood engorgement, compared with non-cut individual mosquitoes?

We did not think about sealing the antenna, and also did not know how to seal the wound. We just reserved the healing time of 2 days based on the observations in our pre-trial. No records were made for orientation speed, probing time, and blood engorgement time, but it was observed that the orientation speed and probing time depended on the number of antenna segments cut compared to uncut individual mosquitoes. In our pre-trials, the localization and detection abilities of mosquitoes with the missing distal antennal segments were almost the same as those of untreated mosquitoes. However, with the increase of missing antennae, the flight activity, orientation speed, and probing time of mosquitoes decreased gradually.

The reviewer's question really triggered our thinking, because the antennae become thinner from the base to the end, and the more antennae segments cut, the larger the wound was. Is the decline of mosquito blood-sucking behavior also related to the increase of wound? Whether wound sealing is needed to promote healing is something to consider. Thank the reviewer for the questions. We have also added these thoughts to the discussion section.

3. Provide more details about human volunteers due to the variation by individual.

Since the variable in this test is the antennae length of mosquitoes, we need to select volunteers as long as they meet the screening criteria and can follow the requirements strictly. A female teacher (33 years old, married and childless) with 165cm height and 51kg weight was finally selected. She claimed that she did not smoke or drink, and the attraction to mosquitoes is moderate. Because of the interesting of this study and could tolerate mosquito bites. She decided to volunteer, which she eventually did it.

4. Italic for all scientific name of the species in text and references.

We have italicized all scientific name of the species in text and references.

5. More discussion about the receptors/sensors. There were many studies about the olfactory sensors, also, single sense cell function/role detected by EAG ..

Thanks to the reviewer for reminding us that the paper lacks sufficient discussion. We have discussed related issues in the revised version.

6. There were several publications about the receptors of antenna of Aedes albopictus in China. You need to search for the Chinese literatures and understand where you stand and start. Huang, XJ, et al. 1991. Studies on the ultra-structures of antennal receptors of Aedes albopictus. Chinese Journal of Vector Biology and Control Vol. 2 (suppl.): 29-32. Xue RD. 1991. Studies on the olfactory behavior of Periplaneta americana, Musca domestica, and Aedes albopictus (NSF project summary). Chin J Vector Biol Control. Vol. 2 (suppl):5-9. Also, there is a reviewing article about this subject. Xue RD. 2016. Host-seeking and blood feeding behavior of Aedes albopictus. Technical Bulletin of the Florida Mosquito Control Association, Vol. 10:2-13. You may find this article from the ResearchGate or the Florida Mosquito Control Association website at www.yourfmca.org

Thank the reviewers for providing us with a lot of learning materials, including reminding us to pay attention to Chinese literature. We learned a lot from them and discussed much in the section of “Discussion and Conclusions”.

Reviewer #2: Nice research, good interpretation of data. However, two points: One, this is not frontline research. I have attached a file the stand of the research of today. With the molecular tools we have today, we should use them to get a clear idea where we could interact genetically. The field is so hot these days that it is hard to keep up. However, that is the name of the game. Two, by cutting the antennae short one does cut all sensory system short. To attribute a change of behavior to olfaction may not be correct. One would have to show that mechanoreception has not been altered.

I can see that a more thorough literature review would improve the manuscript. For example, similar research has been conducted with other small animals (e.g., zooplankton, fruitflies) with similar results. This would then give a better platform for their observed data.  major edit

Thanks to the editor's suggestion, the idea of this paper comes from the observation in the laboratory that mosquitoes with incomplete antennae can still successfully suck blood. We guessed that there might be some protective mechanism in the 13 antennal flagellomeres of mosquitoes. We want to know how much antennal flagellomeres loss will affect its sucking behavior, and we want to show you the result of this idea. We've actually done something at the molecular level based on these results, but we've run into some difficulties and haven't made any progress yet. Anyhow, we have rethought a lot in this process and added a lot of content to the "Discussion and Conclusion" section.

---

## [Decision Letter · Decision Letter 1]

1 Mar 2023

PONE-D-22-26785R1Effects of antennal segments defects on blood-sucking behavior in Aedes albopictusPLOS ONE

Dear Dr. Chen,

Thank you for submitting your manuscript to PLOS ONE. After careful consideration, we feel that it has merit but does not fully meet PLOS ONE’s publication criteria as it currently stands. Therefore, we invite you to submit a revised version of the manuscript that addresses the points raised during the review process. Please submit your revised manuscript by Apr 15 2023 11:59PM. If you will need more time than this to complete your revisions, please reply to this message or contact the journal office at plosone@plos.org. Please include the following items when submitting your revised manuscript:A rebuttal letter that responds to each point raised by the academic editor and reviewer(s). You should upload this letter as a separate file labeled 'Response to Reviewers'.A marked-up copy of your manuscript that highlights changes made to the original version. You should upload this as a separate file labeled 'Revised Manuscript with Track Changes'.An unmarked version of your revised paper without tracked changes. You should upload this as a separate file labeled 'Manuscript'.

We look forward to receiving your revised manuscript.

Kind regards,

Jiang-Shiou Hwang, Ph.D.

Academic Editor

PLOS ONE

Reviewers' comments:

Reviewer's Responses to Questions

**Comments to the Author**

1. If the authors have adequately addressed your comments raised in a previous round of review and you feel that this manuscript is now acceptable for publication, you may indicate that here to bypass the “Comments to the Author” section, enter your conflict of interest statement in the “Confidential to Editor” section, and submit your "Accept" recommendation.

Reviewer #2: (No Response)

Reviewer #3: (No Response)

2. Is the manuscript technically sound, and do the data support the conclusions?

Reviewer #2: Yes

Reviewer #3: Partly

3. Has the statistical analysis been performed appropriately and rigorously? 

Reviewer #2: N/A

Reviewer #3: I Don't Know

4. Have the authors made all data underlying the findings in their manuscript fully available?

Reviewer #2: Yes

Reviewer #3: Yes

5. Is the manuscript presented in an intelligible fashion and written in standard English?

Reviewer #2: Yes

Reviewer #3: No

6. Review Comments to the Author

Reviewer #2: So, here my evaluation of the revised manuscript.

All little points are fine, the English is somewhat charming but understandable.

The value of this manuscript is so/so. Very old-fashioned research when compared with the research done on Drosophila, for example.

Here an abstract of similar research:

Functional Interaction Between Drosophila Olfactory Sensory Neurons and Their Support Cells

by Sinisa Prelic et al --- https://doi.org/10.3389/fncel.2021.789086

Insects detect volatile chemicals using antennae, which house a vast variety of olfactory sensory neurons (OSNs) that innervate hair-like structures called sensilla where odor detection takes place. In addition to OSNs, the antenna also hosts various support cell types. These include the triad of trichogen, tormogen, and thecogen support cells that lie adjacent to their respective OSNs. The arrangement of OSN supporting cells occurs stereotypically for all sensilla and is widely conserved in evolution. While insect chemosensory neurons have received considerable attention, little is known about the functional significance of the cells that support them. For instance, it remains unknown whether support cells play an active role in odor detection, or only passively contribute to homeostasis, e.g., by maintaining sensillum lymph composition. To investigate the functional interaction between OSNs and support cells, we used optical and electrophysiological approaches in Drosophila. First, we characterized the distribution of various supporting cells using genetic markers. By means of an ex vivo antennal preparation and genetically-encoded Ca2+ and K+ indicators, we then studied the activation of these auxiliary cells during odor presentation in adult flies. We observed acute responses and distinct differences in Ca2+ and K+ fluxes between support cell types. Finally, we observed alterations in OSN responses upon thecogen cell ablation in mature adults. Upon inducible ablation of thecogen cells, we notice a gain in mechanical responsiveness to mechanical stimulations during single-sensillum recording, but a lack of change to the neuronal resting activity. Taken together, these results demonstrate that support cells play a more active and responsive role during odor processing than previously thought. Our observations thus reveal that support cells functionally interact with OSNs and may be important for the extraordinary ability of insect olfactory systems to dynamically and sensitively discriminate between odors in the turbulent sensory landscape of insect flight.

Reading this abstract we can see that the discussion about "wound healing" is way out considering what the researchers above published.

In this context, the results of the authors are fine as results. One has to start somewhere. However, they have to connect with literature like above to make it clear, that their research is the first step to more targeted research where one would use their step as step one and continue with research as above as the guiding research for further meaningful research.

I realize that this field is very competitive and to get a clear step forward is hard to achieve.

So, based on this, it is your decision.

One, the research is not up to the scientific level of your journal;

or

Two, for a review one would need another round of editing with a clear appreciation of the research done on other species (see literature cited in above).

I would prefer solution TWO. These researchers would gain a lot of new knowledge and would move in the direction of modern olfactory research. However, you may not have the pages for it ( send it to an appropriate journal).

Reviewer #3: The investigation has been done on very obvious and general objective: whoever study basic moth part and feeing mechanisms can expect antenatal segment efects will affect blood sucking behaviour. However, the specific study on mosquito antennal flagellomeres and particularly is an enhancement of nowledge. Which flagellomere may affect their blood-feeding behavior, and through rationally designed behavioral experiments, we found that the 6th and 7th flagellomeres on Aedes albopictus antenna are important in the olfactory detection of host searching. Meanwhile, the morphology and distribution of sensilla on each antenna flagellomere were also analysed and discussed in this paper. Over all It is an interesting study and provided some new findings for further understanding the antenna with sensors /receptors function. However, the results are nothing rather, along the expected line.

My specific comments are as follow:

Abstract: Too general statement has been provided in the introductory sentences in abstract The blood-feeding transmits extremely harmful infections including malaria, yellow fever, dengue fever, and other arboviruses, making mosquitoes one of the most harmful creatures to human health.

Methods:

Authors have neither performed proper sealing of the antenna, nor studied orientation activity of the antenna, this implication some line of future studies should be explained.

Repetition of the scientific name Aedes albopictus

Write ones’ full name then abbreviate form as Ae albopictus

Discussion: There should be some lines of discussion on the host-seeking and blood feeding behaviour as this is a complicated issue, not only the receptors on the antenna, also, other parts of organs, such as proborese , and tarsal, mating status, distance between hosts and mosquitoes, etc. are also important determinant of the behaviour so proper discussion explaining implication of these results on vector transmission, blood sucking an knowledge enhancement behavioural attributes of host seeking blood feeing should be explained.

7. PLOS authors have the option to publish the peer review history of their article (what does this mean?). If published, this will include your full peer review and any attached files.

Reviewer #2: No

Reviewer #3: No

---

## [Author Response · Author response to Decision Letter 1]

17 Apr 2023

We thank the editor and reviewers for their time, feedback, and constructive suggestions. We have addressed the reviewers’ comments point by point below. Changes have been documented in the manuscript in highlight text. We have also added a few citations to the manuscript.

Reviewer #2: So, here my evaluation of the revised manuscript.

All little points are fine, the English is somewhat charming but understandable.

The value of this manuscript is so/so. Very old-fashioned research when compared with the research done on Drosophila, for example.

Here an abstract of similar research:

Functional Interaction Between Drosophila Olfactory Sensory Neurons and Their Support Cells

by Sinisa Prelic et al --- https://doi.org/10.3389/fncel.2021.789086

Insects detect volatile chemicals using antennae, which house a vast variety of olfactory sensory neurons (OSNs) that innervate hair-like structures called sensilla where odor detection takes place. In addition to OSNs, the antenna also hosts various support cell types. These include the triad of trichogen, tormogen, and thecogen support cells that lie adjacent to their respective OSNs. The arrangement of OSN supporting cells occurs stereotypically for all sensilla and is widely conserved in evolution. While insect chemosensory neurons have received considerable attention, little is known about the functional significance of the cells that support them. For instance, it remains unknown whether support cells play an active role in odor detection, or only passively contribute to homeostasis, e.g., by maintaining sensillum lymph composition. To investigate the functional interaction between OSNs and support cells, we used optical and electrophysiological approaches in Drosophila. First, we characterized the distribution of various supporting cells using genetic markers. By means of an ex vivo antennal preparation and genetically-encoded Ca2+ and K+ indicators, we then studied the activation of these auxiliary cells during odor presentation in adult flies. We observed acute responses and distinct differences in Ca2+ and K+ fluxes between support cell types. Finally, we observed alterations in OSN responses upon thecogen cell ablation in mature adults. Upon inducible ablation of thecogen cells, we notice a gain in mechanical responsiveness to mechanical stimulations during single-sensillum recording, but a lack of change to the neuronal resting activity. Taken together, these results demonstrate that support cells play a more active and responsive role during odor processing than previously thought. Our observations thus reveal that support cells functionally interact with OSNs and may be important for the extraordinary ability of insect olfactory systems to dynamically and sensitively discriminate between odors in the turbulent sensory landscape of insect flight.

Reading this abstract we can see that the discussion about "wound healing" is way out considering what the researchers above published.

In this context, the results of the authors are fine as results. One has to start somewhere. However, they have to connect with literature like above to make it clear, that their research is the first step to more targeted research where one would use their step as step one and continue with research as above as the guiding research for further meaningful research.

I realize that this field is very competitive and to get a clear step forward is hard to achieve.

So, based on this, it is your decision.

One, the research is not up to the scientific level of your journal;

or

Two, for a review one would need another round of editing with a clear appreciation of the research done on other species (see literature cited in above).

I would prefer solution TWO. These researchers would gain a lot of new knowledge and would move in the direction of modern olfactory research. However, you may not have the pages for it ( send it to an appropriate journal).

We would like to thank the reviewers for giving us the opportunity to revise our manuscript. We would also like to thank the reviewers for providing us with references. We have read this and other related literatures carefully and learned a lot from them. 1. Learned the potential role of antennal auxiliary cells in the sense of smell in mosquitoes and other insects, which increased our attention to these types of cell in the future; 2. Learned the methods of studying the function of ORNs in antennae ex vivo, which are more advanced, objective and intuitive compared with our methods. The direct imaging of cations in living cells can indeed more intuitively observe the response of different parts of antennae to odors. For the above two knowledge points and some other olfactory research methods, we added descriptions in the discussion section.

In this study, we used a behavioral approach to explore the role of antennal segments in mosquitoes' blood-sucking behavior. Since the behavior itself is affected by many factors, we just wanted to try to eliminate the interference of pain factors, so we took the healing time of the wound into consideration. Although we did not know whether the healing of the wound would affect the mosquito's behavior, we did so, and we talked about it. In contrast, cationic imaging of antennae ex vivo observes physiological responses of cells that may not be affected by pain. But anyway, as the reviewer said, our simple and extensive study is the first step of the follow-up research, Thanks to this process, we have learned a lot. Modern methods of olfactory research are also the direction of our efforts.

Reviewer #3: The investigation has been done on very obvious and general objective: whoever study basic moth part and feeing mechanisms can expect antenatal segment efects will affect blood sucking behaviour. However, the specific study on mosquito antennal flagellomeres and particularly is an enhancement of nowledge. Which flagellomere may affect their blood-feeding behavior, and through rationally designed behavioral experiments, we found that the 6th and 7th flagellomeres on Aedes albopictus antenna are important in the olfactory detection of host searching. Meanwhile, the morphology and distribution of sensilla on each antenna flagellomere were also analysed and discussed in this paper. Over all It is an interesting study and provided some new findings for further understanding the antenna with sensors /receptors function. However, the results are nothing rather, along the expected line.

My specific comments are as follow:

Abstract: Too general statement has been provided in the introductory sentences in abstract The blood-feeding transmits extremely harmful infections including malaria, yellow fever, dengue fever, and other arboviruses, making mosquitoes one of the most harmful creatures to human health.

Thanks to the review's suggestion, we have revised the abstract and marked the changes.

Methods:

Authors have neither performed proper sealing of the antenna, nor studied orientation activity of the antenna, this implication some line of future studies should be explained.

have neither performed proper sealing of the antenna, nor studied orientation activity of the antenna, which is attributed to our limited knowledge reserve. Thank you for the reviewer's prompt. We also learned and thought during the revision process, and discussed in the section of shortcomings and limitations.

Repetition of the scientific name Aedes albopictus

Write ones’ full name then abbreviate form as Ae albopictus

Except the first appearance of the scientific name (Aedes albopictus), we have changed the following full name into an abbreviation form as Ae albopictus.

Discussion: There should be some lines of discussion on the host-seeking and blood feeding behaviour as this is a complicated issue, not only the receptors on the antenna, also, other parts of organs, such as proborese , and tarsal, mating status, distance between hosts and mosquitoes, etc. are also important determinant of the behaviour so proper discussion explaining implication of these results on vector transmission, blood sucking an knowledge enhancement behavioural attributes of host seeking blood feeing should be explained.

According to the suggestions of reviewers, in the discussion section, we have focused on the mosquito sensory system and its role in host location, the factors affecting blood-sucking behavior and the methods of mosquito olfaction research.

---

## [Decision Letter · Decision Letter 2]

13 Jul 2023

PONE-D-22-26785R2Effects of antennal segments defects on blood-sucking behavior in Aedes albopictusPLOS ONE

Dear Dr. Chen,

Thank you for submitting your manuscript to PLOS ONE. After careful consideration, we feel that it has merit but does not fully meet PLOS ONE’s publication criteria as it currently stands. Therefore, we invite you to submit a revised version of the manuscript that addresses the points raised during the review process.

We look forward to receiving your revised manuscript.

Kind regards,

Jiang-Shiou Hwang, Ph.D.

Academic Editor

PLOS ONE

Journal Requirements:

Reviewers' comments:

Reviewer's Responses to Questions

**Comments to the Author**

1. If the authors have adequately addressed your comments raised in a previous round of review and you feel that this manuscript is now acceptable for publication, you may indicate that here to bypass the “Comments to the Author” section, enter your conflict of interest statement in the “Confidential to Editor” section, and submit your "Accept" recommendation.

Reviewer #2: All comments have been addressed

Reviewer #3: All comments have been addressed

2. Is the manuscript technically sound, and do the data support the conclusions?

Reviewer #2: Yes

Reviewer #3: Yes

3. Has the statistical analysis been performed appropriately and rigorously? 

Reviewer #2: Yes

Reviewer #3: Yes

4. Have the authors made all data underlying the findings in their manuscript fully available?

Reviewer #2: Yes

Reviewer #3: Yes

5. Is the manuscript presented in an intelligible fashion and written in standard English?

Reviewer #2: Yes

Reviewer #3: Yes

6. Review Comments to the Author

Reviewer #2: Yes, do catch up with today's olfaction research. Your starting point is fine. However, for progress we need more detailed information that we can compare with results from other insects.

Reviewer #3: Revision has been done . quality of the Ms is better and understandable now. However the manuscript need minor revision before acceptance . Following clarification amendment are required

1. The revised version has an additional author hope the procedure of authorship modification has been followed.

2. The introductory sentences in abstract The blood-feeding transmits extremely harmful infections including malaria, yellow fever, dengue fever, and other arboviruses, making mosquitoes one of the most harmful creatures to human health. Is too general statement and may be deleted from abstract?

3. Introduction: line 21-24: After mating, female mosquitoes.…. blood feeding. Shorten the sentence.

4. line 41: This paper attempts…molecular level; Could be rewritten as This paper combines morphological and behavioural studies to determine the functional regions of olfactory organ of mosquitoes to narrow down the range of research targets at cellular and molecular levels.

5. Material and methods: Line 136-140; The physical characteristics of volunteer could be depicted as pointers like; Age: 33, Sex: female, Height: 165cm.. and so on.

Was there only one participant out of the 23 sign ups?

6. Line 158-159: Grammatical error; Participants were aware of the content and purpose of the study but were unaware of the test order.

7. PLOS authors have the option to publish the peer review history of their article (what does this mean?). If published, this will include your full peer review and any attached files.

Reviewer #2: No

Reviewer #3: No

---

## [Author Response · Author response to Decision Letter 2]

18 Jul 2023

We thank the editor and reviewers for their time, feedback, and constructive suggestions. We have addressed the reviewers’ comments point by point below. Changes have been documented in the manuscript in highlight text. 

Reviewer #2: Yes, do catch up with today's olfaction research. Your starting point is fine. However, for progress we need more detailed information that we can compare with results from other insects.

Thanks for the reviewer's encouraging comment! Under the inspiration of the reviewer, tracking and comparing the results of other insects will be our long-term research plan, as we are conducting molecular research based on this study, real-time information will help us with our future research.

Reviewer #3: Revision has been done. quality of the Ms is better and understandable now. However the manuscript need minor revision before acceptance. Following clarification amendment are required

Thanks for the reviewer's encouraging comment!

1. The revised version has an additional author hope the procedure of authorship modification has been followed.

Thanks for the reviewer's reminder, the added author made important contribution in our revision process, we will add her in accordance with the authorship modification procedures required by the journal.

2. The introductory sentences in abstract The blood-feeding transmits extremely harmful infections including malaria, yellow fever, dengue fever, and other arboviruses, making mosquitoes one of the most harmful creatures to human health. Is too general statement and may be deleted from abstract?

We have deleted the statement from the abstract.

3. Introduction: line 21-24: After mating, female mosquitoes.…. blood feeding. Shorten the sentence.

We have revised the sentence with a shorter statement in Line 25-26: Female mosquitoes require blood meals to complete their oogenesis, during which they release pathogens into the hosts or become infected themselves.

4. line 41: This paper attempts…molecular level; Could be rewritten as This paper combines morphological and behavioural studies to determine the functional regions of olfactory organ of mosquitoes to narrow down the range of research targets at cellular and molecular levels.

We thank the reviewer for the suggestion. We have revised the manuscript accordingly in Line 44-46.

5. Material and methods: Line 136-140; The physical characteristics of volunteer could be depicted as pointers like; Age: 33, Sex: female, Height: 165cm.. and so on.

Was there only one participant out of the 23 sign ups?

The reviewer’s point is well taken. We have depicted the physical characteristics as pointers in Line 138-142.

Yes, among the 23 registrants, there is indeed only one participant. Since this is a preliminary study on the effect of antennal segments on the blood-sucking behavior of female mosquitoes, we hope that the research variable is only the antennal segments of female mosquitoes, so we tried to keep other conditions unchanged, including blood donors. Setting up only one participant and selecting the most suitable participant from multiple registrants was our initial design. However, if more different volunteers had joined the experiment to increase the selectivity of mosquitoes, would different results be obtained? This is also what we are thinking about currently, and it may also be where we need to improve in the future. As for the number of participants, we have mentioned in the " Shortcomings and Limitations" part of the manuscript.

6. Line 158-159: Grammatical error; Participants were aware of the content and purpose of the study but were unaware of the test order.

We are sorry for our grammar errors. We have made corrections in Line 160-161.

---

## [Editor Report · Decision Letter 3]

24 Jul 2023

Effects of antennal segments defects on blood-sucking behavior in Aedes albopictus

PONE-D-22-26785R3

Dear Dr. Chen,

We’re pleased to inform you that your manuscript has been judged scientifically suitable for publication and will be formally accepted for publication once it meets all outstanding technical requirements.

Kind regards,

Jiang-Shiou Hwang, Ph.D.

Academic Editor

PLOS ONE
---

## [Editor Report · Acceptance letter]

3 Aug 2023

PONE-D-22-26785R3 

Effects of antennal segments defects on blood-sucking behavior in *Aedes albopictus*

Dear Dr. Chen:

I'm pleased to inform you that your manuscript has been deemed suitable for publication in PLOS ONE. Congratulations! Your manuscript is now with our production department. 

Kind regards, 

on behalf of

Prof. Jiang-Shiou Hwang 

Academic Editor

PLOS ONE